Hematological and cytochemical characteristics of peripheral blood cells in the argus snakehead (Ophiocephalus argus Cantor)

Wang Xue 1
Wu Zhengjie 1
Wu Shengmei 1
Chen Xianxian 1
Hanif Misbah 2
Zhang Shengzhou szzhang@mail.ahnu.edu.cn 1
1 College of Life Sciences, Anhui Normal University , Wuhu , China
2 College of Life Sciences, Anhui Normal University , Faisalabad , Pakistan
Bienzle Dorothee
Electronic publication date: 2021 Apr 26
Publication date: 2021
Volume: 9
Electronic Location ID: e11234
Received 2020 Jul 30; Accepted 2021 Mar 17
Copyright: ©2021 Wang et al.
Copyright year: 2021
Copyright holder: Wang et al.
License: This is an open access article distributed under the terms of the Creative Commons Attribution License, which permits unrestricted use, distribution, reproduction and adaptation in any medium and for any purpose provided that it is properly attributed. For attribution, the original author(s), title, publication source (PeerJ) and either DOI or URL of the article must be cited.
License URL: https://creativecommons.org/licenses/by/4.0/

Keywords: Cell metrology, Cytochemistry, Microstructure, Ophiocephalus argus, Peripheral blood cells

Funding: Anhui Province KJ2013A126 Natural Science Foundation of Anhui Province 1808085MC82 This work was supported by the Key Program of the Education Bureau of Anhui Province (Grant No. KJ2013A126) and the Natural Science Foundation of Anhui Province (Grant No. 1808085MC82). The funders had no role in study design, data collection and analysis, decision to publish, or preparation of the manuscript.

==============================
Background

The argus snakehead (Ophiocephalus argus Cantor) is a highly nutritious, freshwater, cultured bony fish with a high economic value. The health of the fish is closely related to its blood cells, which are critical for oxygen transport, natural defense, and immunity. We investigated the morphometry, microstructure, and cytochemical characteristics of the peripheral blood cells of O. argus. Our results may provide the basic reference values needed to monitor the health of this fish for large-scale cultivation.

Methods

The number of blood cells in O. argus were counted on a hemocytometer and their size was measured using a micrometer under light microscope. The morphology and classification of the blood cells were studied using Wright’s staining and the cytochemical characteristics were studied using seven chemical stains including peroxidase (POX), Sudan black B (SBB), periodic acid-Schiff (PAS), acid phosphatase (ACP), alkaline phosphatase (ALP), chloroacetic acid AS-D naphthol esterase (AS-D), and α-naphthol acetate esterase (α-NAE).

Results

The peripheral blood cells in O. argus can be classified as erythrocytes, leukocytes, and thrombocytes; of which, females had 2.9597 million/mm3, 88,400/mm3, and 43,600/mm3, respectively, and males had 3.0105 million/mm3, 105,500/mm3, and 34,000/mm3, respectively. Leukocytes consisted of neutrophils, monocytes, large lymphocytes, and small lymphocytes. Eosinophils and basophils were not found. Monocytes were the most numerous leukocytes identified, followed by neutrophils and small lymphocytes, while large lymphocytes were the least frequently identified. Cytochemical staining showed that erythrocytes were only positive for PAS staining. Neutrophils were strongly positive for POX, SBB, and ACP, and positive for all the other cytochemical stains. Monocytes were positive for PAS and α-NAE and were weakly positive for ACP and AS-D staining. Large lymphocytes were positive for PAS and were weakly positive for ALP, AS-D, and α-NAE staining. Small lymphocytes were positive for PAS and weakly positive for AS-D and α-NAE staining. Thrombocytes were positive for PAS and were weakly positive for ACP and AS-D, but negative for the remaining cytochemical stains. The morphology of peripheral blood cells in O. argus was generally similar to that of other fish species, while the cytochemical staining patterns showed clear species specificity.

Introduction

Vertebrate blood consists of plasma and blood cells. Fish blood cells can be classified as erythrocytes, leukocytes, and thrombocytes, and are vital to the body for gas transportation, immune defense, and coagulation, respectively (Chen et al., 2019b; Palmer et al., 2015). Blood cells are very sensitive to changes in internal physiological conditions and external stimuli (Palmer et al., 2015). Variations in blood cell counts, morphology, and various intracellular functional components can be used as direct markers to determine the health status of fish (Fang et al., 2014; Ishikawa, Ranzani-Paiva & Lombardi, 2008).

Traditional Wright’s staining can be used to study the microstructure and classification of fish blood cells (Zhang et al., 2019). Cytochemical staining is commonly used to detect biological macromolecules and the activity of enzymes, and can improve our understanding of the function and physiological state of blood cells and cell lineages (Massar et al., 2012; Shigdar, Harford & Ward, 2009). A number of studies have been conducted on the classification, microstructure, and cytochemical characteristics of peripheral blood cells in fish, especially commercially cultured fish. Tripathi, Latimer & Burnley (2004) determined the hematological reference intervals for koi (Cyprinus carpio), including blood cell morphology, cytochemistry, and ultrastructure. Tavares-Dias & Moraes (2006) described the morphology, cytochemistry, and ultrastructure of thrombocytes and leukocytes in neotropical fish (Brycon orbignyanus). Tavares-Dias (2006) studied the morphology and cytochemistry of erythrocytes, thrombocytes, and leukocytes in four freshwater teleosts: big head carp (Aristichthys nobilis), oscar (Astrootus ocellatus), traíra (Hoplias malabarus), and lambari (Astyanax bimaculatus). Fang et al. (2014) observed the morphology and cytochemistry of peripheral blood cells in Schizothorax prenanti by light and electron microscopy. Bianchi et al. (2014) described the cell morphology and cytochemical characteristics of a native South America catfish (Sorubim lima). Zheng et al. (2016) investigated the ultrastructure and cytochemical properties of the peripheral blood cells of the piebald naked carp (Gymnocypris eckloni) by transmission electron microscopy. Zhang et al. (2019) compared the microstructure and cytochemical characteristics of peripheral blood cells in the crucian carp (Carassius auratus) and grass carp (Ctenpharyngodon idellus). These studies indicated that the major groups and micromorphology of peripheral blood cells in different fish were generally similar. However, there were obvious species-specific differences in the leukocytes, the proportion of various leukocyte types, and the cytochemical characteristics of blood cells.

The argus snakehead (Ophiocephalus argus Cantor) belongs to the family Channidae, perciformes, and is commonly found in the Yangtze River basin and the lower Yellow River in China, as well as in various river systems in Korea, Japan, and Russia (Courtenay & Williams, 2004). O. argus is a very popular, nutritious, economically-farmed freshwater fish (Xiao et al., 2017). In China, the annual production of O. argus is about 510,000 tons (worth approximately 1.6 billion US dollars) (Sagada et al., 2017). The incidence of diseases in O. argus has increased as cultivation has increased (Xu et al., 2017). Fish hematology is one diagnostic tool that can provide useful information in guiding treatment options (Grant, 2015). However, the hematology and cytochemistry of peripheral blood cells in O. argus has not been well-studied. We investigated the number, microstructure, and cytochemical characteristics of peripheral blood cells in O. argus using cell counts, Wright’s staining, and cytochemical methods under a light microscope. Our results may improve the understanding of fish hematology, provide a reference for monitoring the health of artificially-bred O. argus, and provide basic information for further study of the physiology and immunology of this species.

Materials & Methods

Animals and blood smear preparation

Thirty healthy adult O. argus specimens were selected (15 males and 15 females), with an average body length of 30.56–42.78 cm and weight of 960.20–1850.32 g. All fish were obtained from a local Wuhu aquaculture farm between May and September, 2019. Blood samples were collected by caudal vein puncture, and about 2 ml of blood was taken from each fish using a sterile 5 ml syringe and 22 G needle. K2-EDTA was used as an anticoagulant to avoid blood coagulation and blood smears were prepared immediately after blood collection. This work was approved by the ethics committee of Anhui Normal University (permit no. 20190312). Fish handling and sampling techniques were carried out in accordance with standard vertebrate procedures and veterinary practices and in accordance with national and provincial guidelines.

Wright’s staining

The prepared blood smears were air-dried at room temperature and treated with Wright’s reagent according to the guidelines by Hefei Tianda Diagnostic Reagent Co., Ltd. (Hefei, China). The blood smears were placed in a box containing a parallel support frame and little water, stained with Wright’s A solution for 1 min at room temperature, and treated with Wright’s B buffer for 10 min. The samples were rinsed with distilled water several times and were air-dried again at room temperature. Stained blood smears were examined under a light microscope with oil-immersion at 1, 000 × magnification (BM2000, Jiangnan Yongxin Co., Ltd. Nanjing, China).

Cytochemical staining

Cytochemical staining was carried out according to the methods described by Xu (2003), with minor modifications. The prepared blood smears were fixed with formaldehyde vapor for Sudan black B (SBB) and acid phosphatase (ACP) staining, 10% methanol-formaldehyde solution for alkaline phosphatase (ALP) and chloroacetic acid AS-D naphthol esterase (AS-D) staining, and 95% ethanol solution for periodic acid-schiff’s (PAS) and α-naphthol esterase (α-NAE) staining. The specific staining procedures are briefly described as follows:

POX staining: 2 ml 0.1% tetramethylbenzidine ethanol solution (0.1 g tetramethylbenzidine (Sangon, 54827-17-7) dissolved in 100 ml 88% ethanol solution) was mixed with 20 µl sodium nitroferricyanide (Sangon, 13755-38-9) saturated solution and dropped on the smears. Then 0.7 ml dilute H2O2 solution (50 µl 1% H2O2 solution mixed with 10 ml distilled water) was added after the smears had been standing for 1 min, and was air blown until distributed evenly and oxidized for 6 min.

SBB staining: dried blood smears were placed in Sudan Black B (Sangon, 4197-25-5) staining solution for 60 min at 37 °C, then rinsed in 70% ethanol solution and distilled water for 1-2 min.

PAS staining: blood smears were oxidized with 10 mg/ml periodic acid for 18-20 min, and rinsed in distilled water for 2 min. Samples were placed in Schiff’s solution for 60-90 min at 37 °C. After rinsing in a sulfuric acid solution (0.6 g sodium bisulfite (Sangon, 7681-57-4) dissolved in 5 ml 1mol/l hydrochloric acid and 100 ml distilled water) three to four times, the smears were washed with distilled water for 2–3 min.

ACP staining: blood smears were stained with the reaction solution (0.1 g lead nitrate (Xilong, 10099-74-8) and 0.128 g β-sodium glycerophosphate (Sangon, 819-83-0) dissolved in 74 ml distilled water and 12 ml pH 4.7 acetic acid buffer) for 4-4.5 h at 37 °C. Samples were washed with distilled water for 5 min and were immersed in 2% ammonium sulfide solution (Aladdin, 12135-76-1) for 30 min.

ALP staining: the smears were immersed in a substrate incubation solution (10 mg naphthol AS-BI phosphate (Sangon, 1919-91-1), dissolved in 10 ml 0.05 mol/l propanediol buffer, mixed with 10 mg fast blue B salt (Yuanye, 14263-94-6), and then filtered) for 45-60 min at 37 °C, and rinsing in distilled water for 2 min.

AS-D staining: blood smears were stained with the incubation solution (10 mg chloroacetic acid AS-D naphthol (Sangon, 528-66-5), dissolved in 0.5 ml acetone solution, and then mixed in 5 ml distilled water, 5 ml pH 7.5 Veronal acetic acid buffer, and 10 mg fast blue B salt (Yuanye, 14263-94-6) were added). Samples were then incubated for 60-80 min at 37 °C, and washed with distilled water.

α-NAE staining: the smears were placed in the reaction solution (100 ml phosphate buffer mixed with 1 ml 4mg/ml α-naphthol acetate (Sangon, 90-15-3), then 100 mg fast blue B salt was added (Yuanye, 14263-94-6), and filtered by oscillation) and samples were incubated for 45-60 min at 37 °C and washed with distilled water three to four times.

After cytochemical staining, the smears were counterstained with Wright’s reagent for POX and SBB, 20 mg/ml methyl green (Sangon, 7114-03-6) for PAS, ACP, and α-NAE, and 1 mg/ml hematoxylin (Sangon, 517-28-2) for ALP and AS-D.

Evaluation of cytochemical staining results

The results of cytochemical staining were expressed in terms of the intensity of cytochemical reactions (negative reaction (−), weak positive reaction (+), positive reaction (+ +) and strong positive reaction (+ + +)), according to the evaluation method described by Bianchi et al. (2014).

Blood cell counts and measurements

The total number of blood cells was calculated using a hemocytometer under an Olympus BX61 microscope (Tokyo, Japan). The number of erythrocytes (RBC), leukocytes (WBC), and thrombocytes (TC) were calculated according to the proportions of these cells counted on the Wright’s blood smears (total number × the percentage of cells). The percentages of different leukocyte types were calculated after counting 3,000 randomly-selected leukocytes from males and females. The cell sizes (the length and width of various cells and nuclei) were obtained using an ocular micrometer scale by measuring 20 randomly-selected cells for each cell type from male and female specimens. The hemoglobin (Hb), hematoceit (HCT), and erythrocyte sedimentation rate (ESR) were determined according to the methods described previously (Peng et al., 2018). The mean corpuscular volume (MCV), mean corpuscular hemoglobin (MCH), and mean corpuscular hemoglobin concentrations (MCHC) were calculated from RBC, HCT, and Hb according to the formulae below (Gao et al., 2007a): MCV(fl)=HCT/RBC

MCH(pg)=Hb/RBC

MCHC(g/dl)=Hb/HCT

Statistical analysis

The experimental data were represented by mean ± SD. The significant differences in morphometric values among different cell types or between sexes were compared by one-way ANOVA analysis using SPSS 21.0 software (SPSS Inc, Chicago, USA). A P-value less than 0.05 was used to indicate a significant difference, and a P-value less than 0.01 indicated an extremely significant difference.

Results

Classification and counting of peripheral blood cells

The blood smears treated with Wright’s staining were observed under a light microscope with oil-immersion at 1, 000 × magnification. According to the morphology and size of cells and nuclei, nucleo-cytoplasmic ratio, the presence or absence of particles, and tinctorial feature in the cytoplasm, the peripheral blood cells of O. argus could be divided into erythrocytes, leukocytes, and thrombocytes, and the leukocytes could be further subdivided into neutrophils, monocytes, large lymphocytes, and small lymphocytes.

The number of erythrocytes, leukocytes, thrombocytes, and various hematological parameters in O. argus were calculated and shown in Table 1. There was no significant difference in the number of erythrocytes and total leukocytes between sexes (P > 0.05), while the number of thrombocytes in females was significantly higher than that in males (P < 0.05). The number of different leukocytes were also shown in Table 1. Monocytes were the most abundant leukocytes in O. argus, followed by neutrophils and small lymphocytes, and large lymphocytes were the least numerous (one-way ANOVA: F4,11 =354.476. P <  0.01). The number of large lymphocytes and small lymphocytes in females was significantly lower than that in males (P < 0.05). No statistically significant differences in Hb, HCT, ESR, MCV, MCH, and MCHC were found between females and males (P > 0.05).

Table 1 Haematological parameters of female and male argus snakehead.

	Female (N = 15	Males (N = 15)	
Parameters	Mean ± SD	Range	Mean ± SD	Range	
TBC (×106/mm3)	3.09 ± 0.21	2.90−3.30	3.15 ± 0.24	2.85−3.45	
CV of TBC (%)	6.95		7.70		
RBC (×106/mm3)	2.96 ± 0.16	2.78−3.16	3.01 ± 0.24	2.72−3.30	
CV of RBC (%)	5.41		7.97		
Hb (g/dl)	10.53 ± 0.37	10.17−11.03	10.60 ± 0.29	10.30−11.0	
HCT (%)	42.41 ± 2.38	39.45−45.27	43.21 ± 1.77	41.52−45.66	
ESR (mm/h)	1.52 ± 0.22	1.23−1.76	1.49 ± 0.10	1.42−1.63	
MCV (fl)	143.51 ± 13.54	124.84−162.85	144.86 ± 14.15	125.83−167.87	
MCH (pg)	35.72 ± 2.57	32.17−39.69	35.46 ± 3.03	31.21−40.44	
MCHC (g/dl)	24.56 ± 2.59	21.95−29.44	24.73 ± 2.03	22.50−27.56	
WBC (×104/mm3)	8.84 ± 0.47	8.29−9.44	10.55 ± 0.82	9.55−11.56	
CV of WBC (%)	5.33		7.78		
Neutrophil(×104/mm3)	1.82 ± 0.10c	1.71−1.94	2.03 ± 0.16b	1.83−2.22	
Monnocyte (×104/mm3)	5.11 ± 0.27d	4.79−5.46	6.09 ± 0.47c	5.51−6.67	
Large lymphocyte (×104/mm3)	0.60 ± 0.03a*	0.56−0.64	0.77 ± 0.06a*	0.70−0.84	
Small lymphocyte (×104/mm3)	1.31 ± 0.07b*	1.23−1.40	1.67 ± 0.13b*	1.51−1.83	
TC (×104/mm3)	4.36 ± 0.23*	4.09−4.65	3.40 ± 0.27*	3.08−3.73	
CV of TC (%)	5.28		7.94		
Notes.

Each data point represents the mean of three replicates ± SD and the range.

* Significant differences in blood cell counts and haematological parameters between males and females (P < 0.05). Different letters (a, b, c, d) in the same column indicate significant differences between leukoyte types (P <  0.05). Coefficient of Variation (CV) (%) = (Standard Deviation/Mean) ×100%. TBC, total blood cell counts.

The microstructure of peripheral blood cells

Erythrocytes

Mature erythrocytes (Fig. 1A) were oval in shape, with a smooth surface, and contained an ovoid or rod-shaped purple nucleus in the center of the cell, with a light brown or yellowish cytoplasm. The size of the mature erythrocytes and their nuclei are shown in Table 2. The cell length, and nuclear length and width of mature erythrocytes in females were significantly larger than those in males (P < 0.01). A small number of immature erythrocytes (Fig. 1A) were also observed on the blood smears treated with Wright’s staining, and were round and smaller than mature erythrocytes, with round or elliptic, dark purplish-stained nuclei.

Figure 1 Microstructure of peripheral blood cells in argus snakehead (Wright’s staining).

(A) mature erythrocyte (arrow): oval with a long oval nucleus; immature erythrocyte (arrowhead): round, with a round nucleus. (B) neutrophil: globular with a bilobate nucleus, pale blue cytoplasm contained many tiny mauve and reddish particles. (C) neutrophil: round, with an eccentric and oval shape nucleus, cytoplasm contained a large number of fine light purple particles and vacuoles of different sizes. (D) monocyte: oval with a horseshoe-shaped nucleus, cytoplasm contained many small vacuoles. (E) large lymphocyte: round or irregularly round, with many projections on the surface. (F) small lymphocyte: elliptic, with minimal cytoplasm, and some microvilli protuberances at the margin. (G) thrombocyte: spindle-shaped with an oval and mostly centered nucleus. (H) thrombocyte: round with flocculent cytoplasm, often appeared in clusters with multiple cells. Scale bars = 10 µm. The magnification is 1000X.

Table 2 The size of the peripheral blood cells in argus snakehead (Mean ± SD, µm, N = 20).

Cell types	Females	Males	
	Cell length	Cell width	Cell length	Cell width	
Erythrocytes	14.33 ± 1.16**	10.43 ± 1.05	13.11 ± 0.88	10.47 ± 0.98	
(nuclei)	(8.71 ± 0.88**)	(4.60 ± 0.70**)	(7.21 ± 0.90)	(3.55 ± 0.79)	
Neutrophils	16.64 ± 2.16c	14.83 ± 2.45c	15.21 ± 2.21c	13.55 ± 2.16c	
Monocytes	16.60 ± 1.77c	14.61 ± 1.43c	15.51 ± 1.34c	14.29 ± 1.10c	
Large lymphocytes	11.38 ± 2.67b	9.92 ± 2.50b	11.94 ± 2.17b	10.31 ± 2.09b	
Small lymphocytes	6.75 ± 1.36a	6.00 ± 1.30a	6.77 ± 1.13a	6.17 ± 1.06a	
Thrombocytes	19.18 ± 3.19d	6.99 ± 0.74a	14.70 ± 2.50c	6.87 ± 1.03a	
Notes.

** Extremely significant differences between males and females (P < 0.01). Different letters (a, b, c, d) in the same column indicate significant differences among different cells (P < 0.05).

Neutrophils

Neutrophils (Figs. 1B, 2C) were spherical or round in shape, with purplish stained nuclei. The nuclei had a variety of shapes, including bilobate, trilobed, kidney-shaped (or non-bilobed), and bilobed, which were the most frequently observed. The cytoplasm was rich and stained light blue, containing numerous fine mauve and reddish particles.

Figure 2 Cytochemical staining of peripheral blood cells in argus snakehead.

Erythrocytes were positive for PAS (CC), and negative for POX, SBB, ACP, ALP, AS-D and α-NAE staining (AA, BB, DD, EE, FF, GG); neutrophils exhibited strongly positive rection for POX, SBB and ACP (AB, BC, DE), positive for PAS and AS-D (CD, FG), and weakly positive for ALP and α-NAE staining (EF, GH); monocytes showed positive for PAS and α-NAE (CE, GI), and weakly positive for ACP and AS-D (DF, FH), while negative for POX, SBB and ALP staining (AC, BD, EG); large lymphocytes exhibited positive for PAS (CF), and weakly positive for ALP, AS-D and α-NAE (EH, FI, GJ), while negative for POX, SBB and ACP staining (AD, BE, DG); small lymphocytes were positive for PAS (CG), and weakly positive for AS-D and α-NAE (FJ, GK), while negative for POX, SBB, ACP and ALP staining (AE, BF, DH, EI); thrombocytes showed positive rection for PAS (CH), and weakly positive for ACP and AS-D (DI, FK), while negative for POX, SBB, ALP and α-NAE staining (AF, BG, EJ, GL). Scale bars = 10 µm. The magnification is 1,000 ×.

Monocytes

Monocytes (Fig. 1D) were the largest leukocytes in O. argus (Table 2). Most of them were round and oval and a few were irregular. The nuclei were oval, pear-shaped, or horseshoe-shaped, and generally stained purple. The most obvious morphological feature of monocytes was that the cytoplasm contained a large number of vacuoles of different sizes with pseudopodia protuberances at the cell edges.

Large lymphocytes

Lymphocytes could be classified as large and small lymphocytes. Large lymphocytes (Fig. 1E) were generally or irregularly round, with large, oval nuclei on one side of the cells; the purplish nucleus occupied almost the entire cytoplasm. Some large lymphocytes had smooth surfaces and some had small finger-like protuberances on the surface of the cells.

Small lymphocytes

Small lymphocytes (Fig. 1F) were oval in shape with an eccentric, purple, rounded, gapped nucleus that occupied almost the entire cell, and contained a thin rim of pale blue cytoplasm. Small lymphocytes were characterized by numerous microvilli protuberances from the cytoplasmic margins.

Thrombocytes

Thrombocytes showed different shapes in the smears, including round, oval, long ovoid, and spindle. Spindle-shaped thrombocytes (Fig. 1G) were usually isolated and their nuclei were consistent with the shape of the cells, most of which were centered and purplish, and the cytoplasm was nearly colorless. Round thrombocytes (Fig. 1H) usually appeared in clusters with multiple cells, with round, dark purple-stained nuclei, and less cytoplasm was flocculent around the nucleus.

The cytochemical staining characteristics of peripheral blood cells

POX staining

Neutrophils (Fig. 2AB) were strongly positive with blue-black, coarse, and rod-shaped granules in the cytoplasm. The cytoplasms of erythrocytes (Fig. 2AA), monocytes (Fig. 2AC), large lymphocytes (Fig. 2AD), small lymphocytes (Fig. 2AE), and thrombocytes (Fig. 2AF) were light blue without granules; all of these cells were negative.

SBB staining

Neutrophils (Fig. 2BC) were covered with a large number of diffusely-distributed dark black granules in the cytoplasm, which were strongly positive. The cytoplasms of erythrocytes (Fig. 2BB), monocytes (Fig. 2AD), large lymphocytes (Fig. 2BE), small lymphocytes (Fig. 2BF), and thrombocytes (Fig. 2BG) were pale purple without granules; all of these cells were negative.

PAS staining

The cytoplasms of erythrocytes, neutrophils, monocytes, large lymphocytes, small lymphocytes, and thrombocytes (Figs. 2CC–2CH) were purple or dark purple with diffusely granular matter; all of these cells were positive.

ACP staining

Neutrophils (Fig. 2DE) contained a large number of brown-black granules or tablets in the cytoplasm and were strongly positive. Monocytes (Fig. 2DF) and thrombocytes (Fig. 2DI) were weakly positive, with a small number of brown granules in the cytoplasm. The cytoplasms of erythrocytes (Fig. 2DD), large lymphocytes (Fig. 2DG), and small lymphocytes (Fig. 2DH) were pale purple without granules; all of these cells were negative.

ALP staining

Both neutrophils (Fig. 2EF) and large lymphocytes (Fig. 2EH) were weakly positive, with many fine, uniformly-distributed purple granules in the cytoplasm. The cytoplasms of erythrocytes (Fig. 2EE), monocytes (Fig. 2EG), small lymphocytes (Fig. 2EI), and thrombocytes (Fig. 2EJ) were pale yellow without stained granules; all of these cells were negative.

AS-D staining

Neutrophils (Fig. 2FG) were positive, uniformly-distributed red granules. The cytoplasms of monocytes (Fig. 2FH), large lymphocytes (Fig. 2FI), small lymphocytes (Fig. 2FJ), and thrombocytes (Fig. 2FK) were pale red with fine granules; all of these cells were weakly positive. Erythrocytes (Fig. 2FF) were negative with a pale pink cytoplasm.

α-NAE staining

The cytoplasm of monocytes (Fig. 2GI) was filled with gray-black diffused or granular deposits, which were positive. The cytoplasms of neutrophils (Fig. 2GH), large lymphocytes (Fig. 2GJ), and small lymphocytes (Fig. 2GK) were purple with dark brown or purple granules; all of these cells were weakly positive. Erythrocytes (Fig. 2GG) and thrombocytes (Fig. 2GL) were negative; their cytoplasm were purplish without granules.

The cytochemical staining patterns of peripheral blood cells

The cytochemical staining patterns of peripheral blood cells in O. argus are summarized in Table 3. Erythrocytes were positive for PAS and negative for POX, SBB, ACP, ALP, AS-D, and α-NAE staining. Neutrophils exhibited a strongly positive reaction for POX, SBB, and ACP; positive for PAS, and AS-D; and weakly positive for ALP and α-NAE staining. Monocytes were positive for PAS and α-NAE; weakly positive for ACP and AS-D; and negative for POX, SBB, and ALP staining. Large lymphocytes were positive for PAS and weakly positive for ALP, AS-D, and α-NAE, while negative for POX, SBB and ACP staining. Small lymphocytes were positive for PAS, and weakly positive for AS-D and α-NAE, while negative for POX, SBB, ACP, and ALP staining. Thrombocytes showed a positive reaction for PAS and a weakly positive reaction for ACP and AS-D, but were negative for all the other cytochemical staining.

Table 3 Cytochemical staining patterns of peripheral blood cells in argus snakehead.

Cell types	POX	SBB	PAS	ACP	ALP	AS-D	α-NAE	
Erythrocytes	–	–	+ +	–	–	–	–	
Neutrophils	+ + +	+ + +	+ +	+ + +	+	+ +	+	
Monocytes	–	–	+ +	+	–	+	+ +	
Large lymphocytes	–	–	+ +	–	+	+	+	
Small lymphocytes	–	–	+ +	−	–	+	+	
Thrombocytes	–	–	+ +	+	–	+	–	
Notes.

+ + + Strongly positive. + + Positive. + weakly positive. - negative.

Table 4 The values of RBC, Erythrocyte sizes Hb, HCT and MCV in argus snakehead and some other fish species.

Species	RBC (106/mm3)	Erythrocyte sizes (µm)	Hb (g/dl)	HCT(%)	MCV (fl)	References	
Ophiocephalus argus	2.72−3.30	14.33 ± 1. 16 × 10.43 ± 1.05	10.30−11.0	41.52−45.66	125.83−167.87	This study	
Lutjanus guttatus	0.75−3.71	11.04 ± 0.85 (10−13)	7.29−17.03	33.53−71.14	135.66−369.80	Rio-Zaragoza et al. (2011)	
Cichlasoma dimerus	1.68−4.27	9.4−10 × 6.2 −7.3	5.23−8.33	22.5−39.12	70.14−198	Vázquez & Guerrero (2007)	
Acipenser persicus	4.8−7.9		8.60−9.87	29.58−31.72	412.20−621.70	Milad et al. (2016)	
Sorubim cuspicaudus	3.5−14.0	10. 5 ×8.8	10.5 ± 2.3	25.5 ± 5.6		Negrete et al. (2010)	
Betta splendens	1.70−2.21	10.12−15. 26 × 7.37 −12.59	7.1−9.4	31–39	187.28 ± 7.05	Motlagh et al. (2012)	
Horabagrus brachysoma	1.66−2.43		7.2−9.9	21.40−55.61	88.07−335.0	Prasad & Charles (2010)	
Gymnocypris eckloni	1.49−1.78	14.88 ± 0. 76 × 10.02 ± 0.42	5.21−7.93	22.42−36.92	150.98−207.42	Tang et al. (2015)	
Acipenser sinensis	0.85 ± 0.10	17.98 ± 0. 96 × 12.65 ± 0.87				Gao et al. (2007a); Gao et al. (2007b)	
Ctenopharyngodon idella							
Megalobrama	1.76 ± 0.23	12.31 ± 0. 78 × 8.27 ± 0.72			4.08 ± 0.12		
amblycephala	1.53 ± 0.12	13.61 ± 0. 85 × 7.47 ± 0.55			3.06 ± 0.10	Chen et al. (2019a)	
Pelteobagrus fulvidraco	1.41 ± 0.10	12.22 ±  0.92 9.98 ±  0.83			3.02 ±  1.14		
Glyptosternum maculatum		19.39 ± 2. 48 × 15.15 ± 1.91				Zhang et al. (2011)	
Notes.

Notes RBC red blood cell counts

Hb hemoglobin

HCT hematoceit

MCV mean corpuscular volume

Discussion

The number and morphology of erythrocytes in O. argus

Transporting oxygen and carbon dioxide through intracellular hemoglobin is the primary function of the erythrocyte (Minasyan, 2014). Erythrocytes are the predominant blood cell type in the vast majority of fish (Satheeshkumar et al., 2011; Satheeshkumar et al., 2012). The erythrocyte counts were significantly different among various fish (Table 4). The number of erythrocytes in O. argus was comparable to that of spotted rose snapper (Lutjanus guttatus) and cichlid fish (Cichlasoma dimerus), lower than that of the Persian sturgeon (Acipenser persicus) and shovelnose catfish (Sorubim cuspicaudus), and higher than that of the Siamese fighting fish (Betta splendens), Asian sun catfish (Horabagrus brachysoma), and piebald naked carp. Hb and HCT values in O. argus were comparable to those of the spotted rose snapper, and higher than those of cichlid fish, the Persian sturgeon, Siamese fighting fish, and piebald naked carp. The value of MCV in O. argus was comparable to that of the spotted rose snapper, cichlid fish, Asian sun catfish, and piebald naked carp, and lower than that of the Persian sturgeon and Siamese fighting fish. RBC and Hb values are related to the ability of the blood to carry dissolved oxygen (Fazio et al., 2012; Tavares-Dias & Moraes, 2004). RBC, Hb, HCT, and MCV values in fish are related to various factors, including diet, fish body length, age, sex, water temperature, salinity, and living environment (Jawad, Al-Mukhtar & Ahmed, 2004; Kori-Siakpere, Ake & Idoge, 2005; Martins et al., 2011; Milad et al., 2016). Fish with higher values of RBC, Hb, HCT, and MCV were mostly carnivorous, with a wide range of motion and high activity. These results were consistent with previous reports that the carnivorous fish with high activity needed to consume more oxygen and had correspondingly higher values of RBC, Hb, and HCT (Engel & Davis, 1964; Molnár & Tamássy, 1970; Rambhaskar & Rao, 1987). This study showed that there was no significant difference in the values of RBC, Hb, and HCT between males and females, which was consistent with most fish, such as the Persian sturgeon (Milad et al., 2016), Siamese fighting fish (Motlagh et al., 2012), shovelnose catfish (Negrete et al., 2010), and cichlid fish (Vázquez & Guerrero, 2007).

The morphological characteristics of mature erythrocytes of O. argus were similar to those of other fish (Ahmed & Sheikh, 2020; Chen et al., 2019a; Negrete et al., 2010; Vázquez & Guerrero, 2007), which were usually oval in shape with an oval or long-oval nucleus. The size of erythrocytes in O. argus was smaller than in the Chinese sturgeon (Acipenser sinensis) and sisorid catfish (Glyptosternum maculatum), larger than in the Siamese fighting fish and cichlid fish, and similar to the piebald naked carp (Table 4). The erythrocyte size reflects the oxygen transport capacity, and small erythrocytes are better able to transport oxygen (Fang et al., 2014).

We found a small number of immature erythrocytes in the peripheral blood of O. argus, which was consistent with reports in other fish. However, the morphology of immature erythrocytes in O. argus were round with a smaller and mostly-round nucleus, which was somewhat different from that of other fish, such as S. prenanti (Fang et al., 2014), the spotted rose snapper (Rio-Zaragoza et al., 2011), piebald naked carp (Tang et al., 2015), crucian carp, and grass carp (Zhang et al., 2019), whose immature erythrocytes were mostly ovoid or oval in shape, with a larger, elliptic nucleus.

The proportion and morphology of leukocytes in O. argus

Four types of leukocytes, neutrophils, monocytes, large lymphocytes, and small lymphocytes, were found in the peripheral blood of O. argus. There were three types of granulocytes in vertebrates: neutrophils, eosinophils, and basophils (Fang et al., 2014). Almost all bony fish contain neutrophils (or heterophils), but eosinophils and/or basophils exist only in certain species (Zhou et al., 2006). Few fish other than tilapia (Oreochromis niloticus) (Ueda et al., 2001) have both eosinophils and basophils. Most fish have only eosinophils without basophils (Gao et al., 2007b; Milad et al., 2016; Motlagh et al., 2012; Rio-Zaragoza et al., 2011; Vázquez & Guerrero, 2007; Zheng et al., 2016). Few fish have only basophils without eosinophils (Shigdar, Harford & Ward, 2009; Da Silva et al., 2011; Zhang et al., 2019), and some fish have neither eosinophils nor basophils (Chen et al., 2019a; Da Silva et al., 2011; Fang et al., 2014; Tavares-Dias & Moraes, 2004; Zhang et al., 2011). We did not find eosinophils and basophils in the peripheral blood of O. argus.

The percentages of leukocytes were different among various fish species. Lymphocytes were the most abundant leukocytes in most of the fish, such as the South American catfish (Bianchi et al., 2014), turbot (Psetta maxima) (Burrows, Fletcher & Manning, 2001), Siamese fighting fish (Motlagh et al., 2012), shovelnose catfish (Negrete et al., 2010), spotted rose snapper (Rio-Zaragoza et al., 2011), piebald naked carp (Tang et al., 2015), cichlid fish (Vázquez & Guerrero, 2007), and sisorid catfish (Zhang et al., 2011). Neutrophils were most common in some fish, such as the Persian sturgeon (Milad et al., 2016) and S. prenanti (Fang et al., 2014). Monocytes were found to be the most abundant leukocytes in O. argus, similar to the Chinese sturgeon (Gao et al., 2007b).

Fish monocytes are mostly round and oval (though a few are irregular in shape), have phagocytic functions, and are extremely sensitive to environmental variations (Zheng et al., 2016). Mononcytes were the largest leukocytes in O. argus, and were significantly larger than those of S. prenanti (Fang et al., 2014), the spotted rose snapper (Rio-Zaragoza et al., 2011), piebald naked carp (Tang et al., 2015), and neotropical fish (Tavares-Dias & Moraes, 2006). The most distinct morphological characteristic of monocytes in O. argus was that their cytoplasm contained vacuoles of different sizes and pseudopodia protuberances on the cell edge, which was consistent with the monocytes reported in other fish (Tavares-Dias, 2006; Tripathi, Latimer & Burnley, 2004; Zheng et al., 2016) and may be related to their phagocytic function (Zheng et al., 2016).

Lymphocytes belong to agranulocytes and play an important role in both innate and acquired immunity (Shigdar, Harford & Ward, 2009). Compared with the lymphocytes in some other fish (Tavares-Dias, 2006; Zheng et al., 2016), the lymphocytes in O. argus were variable in size and were classified as large lymphocytes and small lymphocytes. Most of the large lymphocytes had small finger-like protuberances on the cell surface. Many microvilli protuberances can also be found in the cytoplasmic edges of small lymphocytes. These features have also been reported in other fish (Burrows, Fletcher & Manning, 2001; Da Silva et al., 2011; Rio-Zaragoza et al., 2011; Vázquez & Guerrero, 2007), and the protuberances on the surface of lymphocytes may be related to the immune function of antigen binding receptor molecules (Scapigliati, 2013).

Neutrophils in O. argus were usually round or spherical with a bilobate nucleus, their cytoplasm contained numerous fine mauve and reddish granules; these morphological characteristics were somewhat different from the reports in some other fish. For instance, the neutrophils of S. prenanti (Fang et al., 2014) were round or irregular-shaped, their nuclei were usually kidney-shaped or trilobed, and their cytoplasm contained a large number of light blue or pink granules. The neutrophils of the shovel-nose catfish (Negrete et al., 2010) had eccentric and round nuclei with light blue granules in the cytoplasm; neutrophils of neotropical fish (Tavares-Dias & Moraes, 2006) were round with an oval-shaped, eccentric nucleus; their cytoplasm contained many purple granules of different sizes. The heterophils of the sisorid catfish (Zhang et al., 2011) were round and regular in shape with kidney-shaped or round nuclei, and the cytoplasm contained pale blue granules.

The morphology and number of thrombocytes in O. argus

Different thrombocyte shapes, including round, oval, oblong, and spindle-shaped, were observed in O. argus, which was consistent with the reports in other fish (Fang et al., 2014; Gao et al., 2007b; Michał, Beata & Wiesław, 2019; Rio-Zaragoza et al., 2011; Zhang et al., 2011; Zheng et al., 2016). The spindle-shaped thrombocytes often existed alone with some vacuoles in the cytoplasm and were related to cell phagocytosis (Nagasawa, Somamoto & Nakao, 2015; Stosik et al., 2002). The round thrombocytes usually appeared in clusters with two to eight cells, which may be related to their hemostatic function (Chen et al., 2019b; Peng et al., 2018).

The number of thrombocytes in O. argus was lower than that of the Siamese fighting fish (Motlagh et al., 2012), shovel-nose catfish (Negrete et al., 2010), and spotted rose snapper (Rio-Zaragoza et al., 2011); higher than that of the Chinese sturgeon (Gao et al., 2007b) and piebald naked carp (Tang et al., 2015); and similar to grass carp, blunt snout bream (Megalobrama amblycephala), yellow catfish (Pelteobagrus fulvidraco) (Chen et al., 2019a; Gao et al., 2007b), and cichlid fish (Vázquez & Guerrero, 2007). The differences in the number of thrombocytes in different species of fish may be related to biotic and abiotic factors and their adaptability to the environment (Pavlidis et al., 2007; Prasad & Charles, 2010).

The cytochemical staining patterns of blood cells in O. argus

We studied the cytochemical characteristics of peripheral blood cells of O. argus using seven staining methods, including POX, SBB, PAS, ACP, ALP, AS-D, and α-NAE for the first time. POX is enzyme-specific to neutrophils in mammals and participates in the defense mechanism of bacterial infection (Tavares-Dias, 2006). SBB and PAS staining were used to detect intracellular glycogen and lipids, which may provide energy for phagocytosis (Ueda et al., 2001). ACP and ALP are lysosomal enzymes involved in phagocytosis and degradation ( Da Silva et al., 2011; Shigdar, Harford & Ward, 2009). AS-D is a specific esterase, which is associated with cellular defense and phagocytic material processing (Shigdar, Harford & Ward, 2009; Tavares-Dias & Moraes, 2007). As a non-specific esterase, α-NAE plays an important role in phagocytosis and antigen presentation (Fang et al., 2014).

The erythrocytes observed in this study were only positive for PAS, which was different from results in other fish, such as S. prenanti (Fang et al., 2014), tilapia (Ueda et al., 2001), crucian carp, grass carp (Zhang et al., 2019), and piebald naked carp (Zheng et al., 2016), whose erythrocytes were negative for PAS. The erythrocytes in O. argus were negative for POX, SBB, ACP, ALP, AS-D, and α-NAE staining, which was consistent with the fish listed above. PAS positivity and SBB negativity indicated that glycogen is the main energy source of erythrocytes in O. argus.

The neutrophils of O. argus exhibited a strongly positive reaction for POX, SBB, and ACP, and were positive for PAS and AS-D, and weakly positive for ALP and α-NAE staining. These results were generally similar to the staining results of the Murray cod (Maccullochella peelii peelii) (Shigdar, Harford & Ward, 2009), but different from reports in other fish. For instance, neutrophils of the fat snook (Centropomus parallelus) (Da Silva et al., 2011) were positive for PAS, SBB, ACP, and NAE, but negative for ALP staining. The American paddlefish (Polyodon spathula) (Petrie-Hanson & Peterman, 2005) and piebald naked carp (Gymnocypris eckloni) (Zheng et al., 2016) were positive for ACP, but negative for SBB staining. The Asian sun catfish (Horabagrus brachysoma) (Prasad & Charles., 2010) was negative for ALP, NAE, and ASD staining. Neutrophils in human and mammals are mainly involved in the phagocytosis and degradation of invading microorganisms (Azevedo & Lunardi, 2003; Rieger & Barreda, 2011; Wang et al., 2019). The strongly positive reactions for POX, SBB, and ACP and the positive reactions for PAS, ACP, AS-D, and α-NAE indicated that the neutrophils of O. argus are similar to those of mammals, which have strong phagocytic and bactericidal abilities.

Monocytes of O. argus were positive for PAS and α-NAE, and weakly positive for ACP and AS-D, but negative for SBB, POX and ALP staining. These results were fairly consistent with those of other fish, such as the S. prenanti (Fang et al., 2014), American paddlefish (Petrie-hanson & Peterman, 2005), Murray cod (Shigdar, Harford & Ward, 2009), tilapia (Ueda et al., 2001), fat snook (Da Silva et al., 2011), and piebald naked carp (Zheng et al., 2016), indicating that fish monocytes have phagocytotic and antigen-presenting functions. Glycogen is the main energy source of phagocytosis and the lack of POX and ALP suggested that fish monocytes had weaker bactericidal abilities.

Lymphocytes of O. argus were positive for PAS, which was different from those of the South American catfish (Bianchi et al., 2014), four freshwater teleosts (Tavares-Dias, 2006), channel catfish (Ictalurus punctactus) (Tavares-Dias & Moraes, 2007), and fat snook (Da Silva et al., 2011), whose lymphocyte were negative for PAS. However, the results were consistent with S. prenanti (Fang et al., 2014), tilapia (Ueda et al., 2001) and piebald naked carp (Zheng et al., 2016), suggesting that there were certain glycogen in the lymphocytes of O. argus.

Thrombocytes were positive for PAS, and weakly positive for ACP and AS-D, while negative for all other cytochemical staining. These results were similar to those of the Murray cod (Shigdar, Harford & Ward, 2009), tilapia (Ueda et al., 2001), and piebald naked carp (Zheng et al., 2016). The thrombocytes of lower vertebrates are functionally similar to platelets in mammals (Michał, Beata & Wiesław, 2019), playing an important role in the process of hemostasis and coagulation (Chen et al., 2018; Ferdous & Scott, 2015; Peng et al., 2018). Studies have investigated whether thrombocytes in some fish have phagocytotic abilities (Da Silva et al., 2011; Michał, Beata & Wiesław, 2019; Shigdar et al., 2007; Shigdar, Harford & Ward, 2009; Tavares-Dias et al., 2007; Zhang et al., 2019). In this study, the thrombocytes of O. argus were positive for PAS, ACP, and AS-D staining, indicating that they may have some phagocytic and material-processing functions.

We conducted a comprehensive study on the morphological metrology, microstructure, and cytochemical characteristics of peripheral blood cells in O. argus for the first time. The results showed that the number of erythrocytes and leukocytes in O. argus was consistent with that of carnivorous fish. The morphology and microstructure of peripheral blood cells in O. argus was similar to those of other fish, while the cytochemical staining patterns have clear species specificity. For example, all of the blood cell types of O. argus were positive for PAS; neutrophils were strongly positive or positive for all the seven kinds of cytochemical staining; monocytes, large lymphocytes, and small lymphocytes were negative for POX and SBB; and thrombocytes were weakly positive for AS-D. Our results may enrich the understanding of the morphology and function of peripheral blood cells of fish, and provide basic data for health assessments in O. argus aquaculture.

Supplemental Information

File S1 Cell length and width of female and male argus snakehead specimens

The length and width of erythrocytes were measured in four types of leukocytes (neutrophils, monocytes, large lymphocytes, and small lymphocytes) and thrombocytes in male and female argus snakehead specimens. The cell length and width of the erythrocyte nucleus was also measured. Twenty cells of each type were randomly selected for measurement, the mean value was taken, and the standard deviation was calculated. All the data were statistically analyzed between males and females and between various cells.

Click here for additional data file.

File S2 Leukocyte numbers in female and male argus snakehead specimens

The percentages of different leukocyte types (neutrophils, monocytes, large lymphocytes, and small lymphocytes) were calculated after counting 3,000 randomly selected leukocytes from female and male specimens. The number of different leukocyte types was calculated by multiplying their percentage by the total number of blood cells. The mean value was taken and the standard deviation was calculated. All the data were statistically analyzed between males and females and between various cells.

Click here for additional data file.

File S3 Blood cell counts of female and male argus snakehead specimens

The total number of blood cells was calculated using a hemocytometer under an Olympus BX61 microscope. The number of erythrocytes, leukocytes, and thrombocytes were calculated according to the proportions of these cells counted in the Wright’s blood smears. The mean value and the standard deviation were calculated. All the data were statistically analyzed between males and females and between various cells.

Click here for additional data file.

File S4 Hb, HCT, ESR, MCV, MCH, and MCHC values in argus snakehead specimens

The Hb, HCT, ESR, MCV, MCH, and MCHC values of individual female and male argus snakehead specimens were measured in triplicate. The mean value was taken after three experiments and the standard deviation was calculated.

Click here for additional data file.

File S5 CV values in argus snakehead specimens.

The CV values of female and male argus snakehead specimens were measured ten times. The mean value was taken after ten experiments, and the standard deviation was calculated.

Click here for additional data file.

File S6 Original parts of figure 2

Original cytochemical staining images.

Click here for additional data file.

Additional Information and Declarations

Competing Interests

Author Contributions

Animal Ethics

Data Availability

The authors declare there are no competing interests.

Xue Wang conceived and designed the experiments, performed the experiments, analyzed the data, prepared figures and/or tables, and approved the final draft.

Zhengjie Wu and Shengmei Wu performed the experiments, authored or reviewed drafts of the paper, and approved the final draft.

Xianxian Chen and Misbah Hanif analyzed the data, authored or reviewed drafts of the paper, and approved the final draft.

Shengzhou Zhang conceived and designed the experiments, authored or reviewed drafts of the paper, and approved the final draft.

The following information was supplied relating to ethical approvals (i.e., approving body and any reference numbers):

The Ethics Committee of Anhui Normal University approved this research (20190312).

The following information was supplied regarding data availability:

The raw measurements are available in the Supplementary Files.

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
