# Peer review of "Hematological and cytochemical characteristics of peripheral blood cells in the argus snakehead (Ophiocephalus argus Cantor)"

_PeerJ, doi:10.7717/peerj.11234_

## Round 0.1 · original submission · Major Revisions

The reviewers have provided constructive feedback on revision of the manuscript, and you are encouraged to consider each point during your revision.

In addition, please note that it would be very valuable if the coefficient of variation of this highly manual method for differential counting was determined. Manual differential counts are inherently variable relative to results from automated hematology analyzers, and the CV of 10 repeated differential counts would be useful.

Further, please consider and discuss that 2-dimensional measurements of cells dried on slides do not necessarily reflect in vivo measurements - some cells are more prone to "flatten" during the drying process, while others remain in a more 3D formation.

Finally, please note that thrombocytes aggregate if the blood sample was insufficiently anti-coagulated.

·

Basic reporting

I found this article interesting, properly structured and well written

Experimental design

This article is an original work about hematological and cytochemical analysis of peripheral blood cells in argus snakehead.
The research question is well defined but in my opinion is incomplete. In general rules, the standard parameters in the vast majority of hematological analysis of peripheral Blood are: hemoglobin (Hb), red blood cell (RBC) count, hematocrit(Hct), white blood cell (WBC) count, thrombocyte count (TC), erythrocyte sedimentation rate (ESR), mean corpuscular hemoglobin (MCH), mean corpuscular hemoglobin concentration (MCHC) and mean corpuscular volume (MCV). In this manuscript some blood parameters are analyzed but there are some that have not been performed . I recommend the authors to complete the analysis to obtain full data to compare with other fish species.

Validity of the findings

I found the discussion too long, I recommend to shorten a little bit. Maybe some comparisons could be summarized in a table and attached as supplementary figure. For example, the number of erythrocytes in different species, the morphological characteristics of mature erythrocytes….
On the other hand, in the discussion when authors discuss the staining results obtained they speculate about of the possible explanation when cells are positives or negatives. In my opinion I have found too many speculations and I believe that authors can look for bibliography that support their results or perform experiments to probe that suggestion.

Additional comments

I have minor comments for the authors:
• In the materials and methods, I do not found commercial references in many reactives and in the microscope. Please add them.
• In the materials and methods, in the section: blood cell count and measurements, authors should include the formula that they used to calculate the number of blood cells.
• In Table 1 you should indicate the N number, mean, SD, indicate and the statistical analysis in the legend. In addition, in my opinion I think you should indicate the count of neutrophil, monocytes, lymphocyte in cell/mm3 as the rest of cells. I have found in some papers for example https://doi.org/10.1080/24750263.2019.1705647 that they usually use the same units and I think it is better to compare between cell types.
• In Figure 1, you should indicate the magnification of the image in the legend.
• In Figure 2, you should indicate the magnification of the image and the scale bar in the legend. Moreover, I found some trouble to know which staining belongs to every image so I recommend to indicate in the edge of the image the initials of the staining used in order to clarify the figure. Also, I found some images blurred for example G2, D2, D4, D6, I recommend to improve them.
• I do not found match with table 3 and Figure 2 images. For example, neutrophils and monocytes have ++ in table3 for α-NAE but in the Figure 2 G2 seems less positive than G3. In the same way, for AS-D neutrophils and thrombocytes have ++ but in figure 2 F6 is more similar to F3 and F2 seems the highest. I don’t know if maybe there is a mistake or these differences are due to the image quality?

·

Basic reporting

Reviewer’s Comments and Suggestions-
1. Criticism- SEM of Blood cells would have given detailed information of the morphology.
2. Suggestion- SEM of blood cells should be done.
3. Introduction on economic importance (with reference) of the fish and the relevance to the current study is not emphasized.
4. Comments on language or grammar issue- The language of the manuscript is clear and reasonably good. However, grammar regarding sentence construction and use of words need some improvement as suggested by the reviewer.

Corrections-
Introduction-
Line 63-7A: A few more references as detailed below should be included. B. Massar, Sudip Dey, R. Barua and K Dutta (2012): Microscopy and Microanalysis of hematological parameters in common carp, Cyprinus carpio L inhibiting a polluted lake in North East India, Micros, Microscopy and microanalysis,18; (1077-1087)

Lines 106-107: Check the English grammar. (“smears should be prepared ………………”) is to be corrected.

Line 117: Mention the make and model of the light microscope.
Line 128: ……………min at 370C, “then” …is the correct grammar.
Line 149: “After rinsing in distilled water for 3 times….” Change into “After three times rinsing….”
Lines 155-56: “After rinsing in distilled water ……” may be changed as “After 3 minutes rinsing in distilled water……”
Line 162: The sentence may be changed into “After 3 times rinsing………”
Lines 186-188: Scanning electron microscope may be used to have detailed information on morphology and size of cells to improve the standard of the manuscript.
Line 265- “………showed weakly positive staining…………”
Line 276: “Neutrophils exhibited strongly positive reaction for……….”
Line 278: “……………weakly positive reaction for……….”
Line 282: “……..showed positive reaction….”
Line 419: “……..strongly positive reaction…..”
Line 448: “……………positive reactions………..”
Line 458:……………….”showed” may be replaced with were……
Line 466: replace “exhibited” by “were”

Experimental design

Adequate. However, SEM study may improve the standard of the manuscript.

Suggestion: Few SEM study may be done.

Validity of the findings

Findings are valid and satisfactory except SEM study.

Suggestion: Few SEM study may be done.

Additional comments

Importance of the issue-

1. Aims and objectives are clear.
2. The manuscript is written clearly with nice micrograph and relevant tables
3. References are adequate and up-to-date.

Strengths as well as weakness of the manuscript-

The authors made a nice attempt and deserve appreciation for their extensive data. The manuscript is clearly written in professional language. If there is weakness, it is in the absence of SEM works. The SEM analysis should be done or at least the need for Scanning electron Microscopy (SEM) should be emphasized in this type of work.
If SEM analysis cannot be done, the reason(s) for the same should be mentioned clearly before ACCEPTANCE.

---

## Round 0.2 · Minor Revisions

The incorporation of the reviewers' comments has improved the manuscript. Please note several outstanding concerns from reviewer 1. Please also note that there are grammatical errors throughout the manuscript, such as outlined in Table 1 (attached). Please indicate how "CV" in Table 1 was calculated.

·

Basic reporting

I found this article interesting, properly structured and well written

Experimental design

This article is an original work about hematological and cytochemical analysis of peripheral blood cells in argus snakehead.The research question is well defined and the methods are described with suficient detail.

Validity of the findings

The authors have addressed all of the corrections suggested, and now the manuscript has substantially improved

Additional comments

The authors have addressed all of the corrections suggested, and now the manuscript has substantially improved. However, there are some issues that it should be corrected that I detailed bellow:
• As the table 3 has been changed now in the discussion there are some sentences that are not update such as:
- Line 768” The strongly positive reactions for POX and SBB” and they should add ACP.
-Line794” were positive for PAS and AS-D staining” and they should add ACP.
• In the table 4, I recommend to include in the legend the meaning of the acronyms (Hb, HCT and MCV ) to clarify the information.
• I found some mistakes that I detail bellow:
-Line 535,541,743: rection, I think is reaction please change it.
-Line 739:whoes , I think is whose please change it.

---

## Round 0.3 · Minor Revisions

Thank you for addressing reviewer 1's points.
Regarding calculation of the CV: I am quite familiar with the formula, but it is unclear what numbers were used to derive at CVs of 9.34 and 8.46. Mean and SD of what parameter does this pertain to? Please provide the raw data as a supplemental file.

---

## Round 0.4 · Minor Revisions

I appreciate that the authors have included a definition of CV in the revised manuscript. However, it is not clear whether "TBC" consists of all erythrocytes, thrombocytes and leukocytes, or only leukocytes. Manual cell counts are highly error prone, and it is important to add information on how much variation was observed from count to count. The red cell mass is well represented by Hgb concentration, but the leukocyte concentration is manually determined and CVs should be provided.

The information in Suppl file 5 is uninformative since the header for each set of numbers is the same. This information needs to be self-explanatory with regard to what cells are being counted.

---

## Round 0.5 · Minor Revisions

Thank you for clarifying the latest concerns. The entire manuscript will require careful attention to English grammar prior to publication. You can consult with colleagues or use a third-party editing service.

In addition, the supplementary tables are not self-explanatory

---

## Round 0.6 · Minor Revisions

I appreciate that a review of the manuscript by an editing service has improved the grammar. However, English language use remains unclear or grammatically incorrect in many sections of the manuscript. For example:

L49 "Blood is extremely important in fish and other vertebrates." - that is obvious and should not be stated.

L26 "blood count board" and L162 "Neubauer counter board" should be "Neubauer counting chamber" or "hemocytometer".

L110 "The blood smears were placed in a wet box," - what is a wet box? A sealed container containing a moist paper towel?

L113: Were the dried blood smears covered with a cover slip?

L117: How do you know they were improvements? Should use "modifications".

L130: I don't think you mean "periodate" but maybe periodic acid?

There are many other inaccurate or erroneous terms.

A PeerJ copyeditor can recheck the manuscript and supplemental files but that service only addresses the language quality. Technical usage should be corrected by the authors first.

---

## Round 0.7 · accepted · Accept

The grammar is the revised manuscript is improved.